# Comparison of the Ecological Traits and Boring Densities of *Aromia bungii* (Faldermann, 1835) (Coleoptera: Cerambycidae) in Two Host Tree Species

**DOI:** 10.3390/insects13020151

**Published:** 2022-01-30

**Authors:** Tadahisa Urano, Hisatomo Taki, Etsuko Shoda-Kagaya

**Affiliations:** 1Kansai Research Center, Forestry and Forest Products Research Institute, 68 Nagaikyutaro, Momoyama, Fushimi, Kyoto 612-0855, Japan; 2Forestry and Forest Products Research Institute, Matsunosato 1, Tsukuba 305-8687, Japan; htaki@affrc.go.jp (H.T.); eteshoda@affrc.go.jp (E.S.-K.)

**Keywords:** *Cerasus* × *yedoensis* ‘Somei-yoshino’, elytral length, emergence trends, lifespan, lifetime fecundity, *Prunus persica*

## Abstract

**Simple Summary:**

The red-necked longhorn beetle *Aromia bungii* is an invasive, wood-boring pest that infests Rosaceae trees. Its main host species in Japan are flowering cherry (*Cerasus* × *yedoensis* ‘Somei-yoshino’) and peach (*Prunus persica*) trees. We evaluated which species is more vulnerable to *A. bungii* by investigating the ecological traits of emerging adults and the boring density of larvae and pupae. The number of emerging adults per m^3^ of *P. persica* trunks was 10-times greater than from *C*. × *yedoensis*, and the numbers of grown larvae and pupae in the trunk were higher in *P. persica* logs. The number of eggs laid over the lifetime of female adults was larger for insects that emerged from *P. persica*. Body size, sex ratios, and adult life spans did not differ between the two host trees. This study elucidated that peach trees are more suitable hosts than cherry trees for *A. bungii* larvae. Although cherry trees, primarily *C*. × *yedoensis*, which are currently grown as street or ornamental trees in Japan, have been more severely affected by *A. bungii* to date, the greater risk in the long term is to *P. persica*, an agricultural species in the main producing areas surrounding the Kanto region.

**Abstract:**

We investigated the ecological traits of emerging adults and the boring density in *Aromia bungii*-infested flowering cherry (*Cerasus* × *yedoensis* ‘Somei-yoshino’) and peach (*Prunus persica*) trees to evaluate their suitability as food resources for *A. bungii*, and their vulnerability to infestation. The number of adults per m^3^ that emerged from *P. persica* was 10-times larger than from *C*. × *yedoensis*, and the numbers of emergence holes, entrance holes, and pupal chambers were also larger in *P. persica* logs. The lifetime fecundity of adults that emerged from *P. persica* was also higher. Elytral length, sex ratios, and adult lifespans did not differ between the two host trees. Our results indicate that peach trees provide more suitable conditions than do flowering cherry trees for *A. bungii* larvae. Although flowering cherry trees, primarily *C*. × *yedoensis*, which are currently grown as street or ornamental trees in Japan, have been more severely affected by *A. bungii* to date, the greater risk in the long term is to *P. persica*, an agricultural species in the main producing areas surrounding the Kanto region.

## 1. Introduction

The red-necked longhorn beetle (RLB) *Aromia bungii* (Faldermann, 1835) (Coleoptera: Cerambycidae) is an invasive wood-boring pest that infests trees in the family Rosaceae. The species is native to China, the Korean peninsula, the Russian far east, Mongolia, and Vietnam, but has become invasive in Germany, Italy, and Japan since 2008–2011 [1,2,3]. Adults emerge in June and July, mate immediately, and initiate oviposition in the subsequent days [4]. Eggs are laid separately in the crevices of the outer bark and hatched larvae feed on the inner bark. Full-grown larvae bore pupal chambers in the sapwood where they overwinter in diapause, and then pupate and emerge the following year. In Japan, the typical timeframe between the egg stage and adult emergence is 2 years [1]. Affected trees are attacked for several years before succumbing

In China, which is within the native range of *A. bungii*, the primary host species are *Prunus salicina* Lindl., *P. mume* (Sieb.) Sieb. et Zucc., *P. persica* (L.) Batsch, and *P. armeniaca* L. [5,6,7,8], with most of the damage being incurred by *P. persica* (peach) [4,9]. This is a result of the high planting densities of peach, which is attributable to its importance as both an agricultural and horticultural species in China [1].

The first *A. bungii* infestation in Japan was recorded in 2012, and damage to cherry trees (mostly *Cerasus* × *yedoensis* (Matsum.) Masam. et Suzuki ‘Somei-yoshino’), *P. mume*, *P. persica*, and *P. salicina* has been documented in 12 prefectures in the Kanto, Chubu, Kinki, and Shikoku regions. In Kanto (including Tokyo, Saitama, Gunma, Tochigi, Ibaraki, and Kanagawa Prefectures), the most heavily impacted area in Japan, most damage has been observed in *C.* × *yedoensis* [10,11]. Because flowering cherry trees have great cultural significance in Japan and are iconic symbols in several scenic locations and historic sites, the impacts of *A. bungii* infestations also extend to tourism and non-economic sectors.

Damage to *P. persica* and *P. salicina* in orchards was observed in Tochigi Prefecture in 2017 [12]. In Tokushima Prefecture (Shikoku), the damage to *P. persica* orchard trees was immediately noticeable [13]. Although *P. persica* appears to be less severely affected than *C*. × *yedoensis* in Japan to date, the former has sustained more damage in areas where both species are simultaneously affected [13]. Most *P. persica* orchard trees die within 2–3 years of oviposition by *A. bungii*. The emergence of adults begins 1 year after large amounts of frass become visible on the trunk as a result of the activity of mature larvae. Visual symptoms of decay also become more obvious in host trees at the time of adult emergence [14]. No studies have explored the time required for *A. bungii* to kill *C*. × *yedoensis*. However, the process is thought to take several years as affected *C*. × *yedoensis* trees are generally larger than *P. persica* trees.

As previously indicated, both *P. persica*, which is an agricultural species, and *C*. × *yedoensis*, which is a street or ornamental tree, are currently affected in the same area. If these species differ with respect to the amount of damage they sustain, the differences may indicate host preference of ovipositing adults [15] as well as differences in conditions for larval development in the trunk [16]. Understanding the differences between the two host species is important for predicting future impacts and identifying effective control methods. We collected adults of *A. bungii* that emerged from infested *C*. × *yedoensis* and *P. persica* logs stored in outdoor cages, and reared them in the laboratory. We compared emergence trends, the numbers of emerging adults, sex ratios, elytral length, lifespan, and the lifetime fecundity of females between the two host trees. The numbers of emergence holes, entrance holes, and pupal chambers were also compared by dissecting the logs. Based on these quantitative data, we clarify differences between the two tree species with respect to their susceptibility to *A. bungii*, which are poorly understood, and discuss potential future trends in Japan.

## 2. Materials and Methods

### 2.1. Sample Log Collection

Adults of *A. bungii* were collected from infested trees felled at two locations in the Kanto region, Japan.

Logs of *C*. × *yedoensis* were collected from Soka City, in Saitama Prefecture (35°84′ N, 139°83′ E). Damage resulting from *A. bungii* was initially observed in 2013 in a row of trees planted along a canal (Kasai Yosui) in 1978 [10]. Most of the dead or damaged trees were felled and cut into chips. Although damage to the trees along the canal is currently diminishing due to control efforts, *A. bungii* has dispersed to other trees planted in the surrounding area near private homes, factories, schools, and other sites, and the insect continues to spread. Twenty-one infested *C*. × *yedoensis* trees (5–10 m tall) at this site were felled and cut into logs in June and October 2017 and June 2018.

*Prunus persica* logs were collected from Sano City, in Tochigi Prefecture (36°28′ N, 139°55′ E). Damaged *P. persica* and *P. salicina* trees have been observed in the orchards since 2017, and 498 of 2603 trees (19.1%) were infested in 2018 [17]. Eleven infested *P. persica* trees (10–15 years old, 2.5–3.0 m tall) at this site were felled and cut into logs in May 2018.

All logs were placed in outdoor cages (2 m wide × 2 m deep × 2.55 m high) at the Forestry and Forest Products Research Institute (Tsukuba City, Ibaraki Prefecture). Logs of the two species were kept in separate cages (one for *P. persica* and three for *C*. × *yedoensis*). The diameter and length of each log were measured.

### 2.2. Collection and Rearing of A. bungii

We monitored the cages every morning between late May and early August, and collected all emerged adults. Each female was paired with a male in a plastic cup (13-cm diameter × 5-cm high) for approximately 24 h within one day of collection. Adults were then reared separately in the same-sized plastic cups lined with a filter paper (Advantec No. 2, Advantec, Tokyo, Japan; 9-cm diameter), and provided with a cotton wool ball soaked with 25% diluted honey as food. For females, based on their habit under natural conditions of inserting their ovipositor into a crack in the outer bark for oviposition, a piece of cardboard (3 × 3 cm) was placed on the bottom of the cup, allowing the female to oviposit between the cardboard and the filter paper. The number of eggs laid by each female was counted every 2–4 days. All adults were reared at 25 °C under a 16 L:8 D photoperiod until death, and we recorded lifespan and female lifetime fecundity (i.e., the number of eggs laid over the full lifetime). Elytral length (i.e., the length of the left elytra) was measured immediately after death using a digital caliper (Mitutoyo IP67, Mitutoyo Co., Kawasaki, Japan), and was used as a proxy of adult size.

### 2.3. Dissection of Sample Logs

Between November 2018 and January 2019, five logs of each species were selected and dissected following the cessation of adult emergence in the outdoor cages. Large logs (more than 30 cm in diameter) of *C*. × *yedoensis* were not selected owing to the limited processing capacity of the wood splitting machine. *Aromia bungii* larvae feed and develop under the bark and bore into the sapwood, where they create pupal chambers in which they overwinter in diapause. An entrance hole on the surface of sapwood indicates the presence of a grown larva in the log. Prior to overwintering, full-grown larvae bore through the outer bark from the inside, creating holes to facilitate emergence. Emerging adults do not bore emergence holes directly from the pupal chamber but, rather, through the galleries, and emerge through the entrance and emergence holes.

After measuring the length and diameter of each log, we counted the number of emergence holes on the log surface. The logs were then debarked using hatchets and knives, and the numbers of entrance holes in the sapwood were counted. Sample logs were then split with a wood-splitting machine (PS42NML, Shingu Shoko Ltd., Otaru, Japan), and the numbers of pupal chambers, along with the numbers of live and dead individuals in the pupal chambers, were recorded.

### 2.4. Statistical Analyses

To examine the effects of tree species on the size and volume of sample logs and the lifetime fecundity of female adults, we constructed generalized linear models (GLMs) with species as a fixed factor and lifetime fecundity and the diameter and volume of the logs as the response variables. Akaike’s Information Criterion (AIC) values were used to compare the model including tree species to a null model. The sizes and volumes of sample logs were assumed to follow a Gamma distribution, whereas lifetime fecundity was assumed to follow a Poisson distribution.

To test the effects of tree species and *A. bungii* sex on the lifespan and elytral length of adults, GLMs were constructed using species and sex as fixed factors and lifespan and elytral length as the response variables. AIC values were again used to compare among models that included one or both fixed factors, an interaction model, and a null model. The response variables were assumed to follow a Gamma distribution.

To examine the effects of tree species on the volume of sample logs and the numbers of emergence holes, entrance holes, and pupal chambers observed during log dissection, GLMs were constructed using species as a fixed factor and log volume and the numbers of emergence holes, entrance holes, and pupal chambers as the response variables. AIC values were used to compare between the model including tree species and a null model. The volume of sample logs was assumed to follow a Gamma distribution and the numbers of emergence holes, entrance holes, and pupal chambers were assumed to follow a Poisson distribution.

Pearson’s correlation coefficients were calculated to explore the relationship between insect lifespan and elytral length by tree species, and the relationships of lifetime fecundity to lifespan and the elytral length of females. 

All statistical analyses were performed in R version 4.0.5 [18].

## 3. Results

### 3.1. Comparison of Log Size between Tree Species

The diameters and volumes of the sample logs are shown in Figure 1. The average height and diameter of *C*. × *yedoensis* trees were larger than those of *P. persica*, because *C*. × *yedoensis* had been cultivated on roadsides and in school yards and parks, whereas the *P. persica* were orchard trees. Logs of *C*. × *yedoensis* generally had large diameters but were cut to short lengths to facilitate hauling. Conversely, *P. persica* logs tended to be longer owing to their smaller diameters. Diameter, length, and volume of sample logs are presented in Appendix A.

The AIC values for the two models that included the effect of tree species (diameter = 895.71, volume = −725.86) were smaller than those of the corresponding null models (diameter = 957.24, volume = −672.22), indicating that *C*. × *yedoensis* logs had significantly larger diameters and volumes.

### 3.2. The Number and Sex Ratio of Emerging Adults

The numbers and sex ratios of adults that emerged from the two tree species are shown in Table 1. The total volume of *C*. × *yedoensis* logs was 5.5-times greater than that of *P. persica* logs; however, larger numbers of adults, both male and female, emerged from the *P. persica* logs. The number of adults of both sexes that emerged per m^3^ of *P. persica* logs was 10-times larger than the number that emerged from *C*. × *yedoensis* logs. The sex ratio of adults, expressed as the proportion of males, was nearly 1:1 (0.52) in both tree species. 

### 3.3. Emergence Trends of Adults

Emergence trends for adults by sex and tree species are shown in Figure 2. Adults of both sexes emerged from *C*. × *yedoensis* logs between mid-June and late July. Adult males emerged from *P. persica* logs between early June and early July, and adult females between mid-June and early July. The date of 50% accumulation of adults emerging from *P. persica* occurred two days earlier for males and one day earlier for females. The emergence period was shorter in *P. persica* than in *C*. × *yedoensis*. In *C*. × *yedoensis*, the first day of emergence was the same for both sexes, the date of 50% accumulation occurred 2 days earlier for males than for females, and the last day of emergence occurred 3 days earlier for males than for females. In *P. persica*, the first day of emergence occurred 11 days earlier for males than for females, the date of 50% accumulation of males occurred 3 days earlier, and the last day of emergence for males occurred 9 days earlier.

### 3.4. Elytral Length of Adults

The elytral lengths of adults by sex and tree species are shown in Figure 3. The AIC values of the GLM including tree species and sex of adults as fixed factors were 1341.0 (largest) for the model including only tree species, 1328.9 (smallest) for the model including only sex, 1330.9 for the model including both tree species and sex, 1330.9 for the model including both fixed factors and their interaction, and 1339.1 for the null model. Therefore, sex had a significant effect on elytral length, but tree species did not. Elytral lengths varied more among males than among females: the difference between the maximum and minimum individual sizes was 1.92 times for males and 1.63 times for females. Data of elytral lengths are presented in Appendix A.

### 3.5. Adult Lifespan

Adult lifespans by tree species are shown in Figure 4. The AIC values of GLMs were 1613.3 (largest) for the model including only tree species, 1586.8 (smallest) for the model including only sex, 1587.2 for the model including both predictors, 1589.0 for the model including both fixed predictors and their interaction, and 1611.5 for the null model. Therefore, lifespan differed between males and females, but was not affected by the host species.

Females had a longer average lifespan, but the longest-lived individuals in both tree species were males. Pearson’s correlation coefficients indicated no significant relationship between elytral length and lifespan for either tree species or sex (males from *C*. × *yedoensis*: r = −0.088, *p* = 0.62; females from *C.* × *yedoensis*: r = −0.07, *p* = 0.688; males from *P. persica*: r = −0.017, *p* = 0.889; females from *P. persica*: r = −0.138, *p* = 0.396). Data of lifespan are presented in Appendix A.

### 3.6. Lifetime Fecundity

The lifetime fecundity of female adults by host species is shown in Figure 5. The number of eggs laid per individual varied substantially (minimum = 10, maximum = 1142). The AIC values of the GLM including tree species (10,626) was smaller than that of the null model (10,766). The lifetime fecundity of females that emerged from *P. persica* was significantly greater than that of females that emerged from *C*. × *yedoensis*. Data of lifetime fecundity are presented in Appendix A.

The Pearson’s correlation coefficients between elytral length and lifetime fecundity were not significant (females from *C*. × *yedoensis*: r = −0.309, *p* = 0.05; females from *P. persica*: r = −0.198, *p* = 0.295). Correlation coefficients between lifespan and lifetime fecundity were significant for females that emerged from *P. persica* (r = 0.608, *p* < 0.001), but not *C*. × *yedoensis* (r = 0.199, *p* = 0.217).

### 3.7. Dissection of Sample Logs

The results of sample-log dissection are shown in Figure 6. The AIC values were larger for the GLM that included tree species (−77.028) than for the null model (–78.813), indicating that there was no difference in the volume of sample logs between the two tree species. The AIC values of the GLMs for the numbers of emergence holes, entrance holes, and pupal chambers were smaller for models that included the effect of tree species (emergence holes = 43.073, entrance holes = 53.422, pupal chambers = 55.223) than for the corresponding null models (emergence holes = 69.566, entrance holes = 124.56, pupal chambers = 148.94). *Prunus persica* logs had larger numbers of larvae boring into the sapwood, more pupal chambers, and more emerging adults than did the *C*. × *yedoensis* logs. The number of pupal chambers per wood volume (m^3^) was 218.79 ± 78.56 (mean ± SD) for *C. × yedoensis* and 2004.00 ± 986.85 for *P. persica*. Diameter, length, and volume of sample logs and numbers of emergence holes, entrance holes, pupal chambers, larvae, and adults observed during log dissection are presented in Appendix A.

Although most pupal chambers in the dissected logs were empty (the adults having already emerged), one alive larva and one dead adult were collected from *C.* × *yedoensis*, and 18 living larvae, 9 dead larvae, and 6 dead adults were collected from *P. persica*. The fresh weight of the 16 living larvae was 984.29 ± 669.14 mg (mean ± SD) with a maximum of 2453.7 mg and a minimum of 79.13 mg. We noted that immature larvae also bored into sapwood. Fresh weight of living larvae in the dissected logs are presented in Appendix A.

Due to the high larval boring densities in *P. persica* logs, many cracks were observed in the bark and most of the inner bark (cambium) was consumed. The density of larval galleries under the bark of *C. × yedoensis* logs was much lower than that of *P. persica*. The number of adults that actually emerged from *P. persica* was closer to the number of pupal chambers than the number of emergence holes, because the cracks in the bark made it difficult to locate all emergence holes.

## 4. Discussion

The sex ratio (proportion of males) of emerging adults was 0.52 for both tree species, which is nearly 1:1. Similar sex ratios (0.42 [19], 0.48 [6], and 0.57 [7,20]) have been reported in studies from China.

The emergence period of adults in this study occurred earlier than reported by other studies conducted in Japan, in which emergence periods were late June–early July [11], late June–mid-July [10], and late June–early August [13]. This is assumed to be a result of higher-than-average temperatures between March and June 2018 [21]. Males generally emerged earlier, as previously reported for this species [19,22] and several other cerambycid beetles [23]. Emergence trends did not differ between tree species; we observed only a 1–2-day difference in the date of 50% cumulative emergence for both sexes. The emergence period was shorter in *P. persica* than in *C*. × *yedoensis*. All the *P. persica* logs were collected from the same orchard and placed into the outdoor cage on the same day, whereas *C*. × *yedoensis* logs were collected from multiple sites in Soka City over three collection periods between June 2017 and June 2018. As such, we assume that there was some variation in the developmental stages of *A. bungii* in the *C*. × *yedoensis* logs.

Reports on adult lifespans vary from a minimum of approximately 10 days to a maximum of 178 days [4,20,22]. Lifespan is thought to vary substantially depending on conditions, such as food availability.

There were significant correlations in adult females between body length and the number of ovarioles, and body length and lifetime fecundity [23]. Significant correlations between body size and lifetime fecundity have been observed in many other cerambycid beetles [24,25,26,27,28,29,30]. We found no significant correlation between body size and the lifetime fecundity of female adults that emerged from either tree species. We observed large differences in egg production among individuals, from a minimum of 10 to a maximum of 1142, as well as substantial variation within all size (elytral length) classes. Individual differences in lifetime fecundity (minimum–maximum of egg production) have also been reported by Liu (9–375) [6] and Yu and Gao (112–362) [9]. The non-significant correlation we observed is likely attributable to these differences, but the reason for the differences should be addressed in future research

Significant correlations between lifespan and lifetime fecundity have also been documented for other cerambycid species [26,29,30]. Significant correlations between the lifespan and lifetime fecundity of females that emerged from *P. persica* are assumed to be attributable to the tendency of females to continue laying eggs throughout most of their lifetime.

Log-dissection results indicated that, despite the nearly identical log volumes between tree species, the number of emergence holes was 5.6-times greater in *P. persica* than in *C*. × *yedoensis* while the number of entrance holes was 5.9-times greater and the number of pupal chambers was 7.8-times greater. Furthermore, the number of individuals emerging per m^3^ of *P. persica* logs was approximately 10-times higher than for *C*. × *yedoensis*. Increased larval density in the trunk entails decreases in resource availability per individual, leading to decreases in the average size of emerged adults. However, we observed no difference in the elytral length and lifespan of the adults that emerged from the two tree species, and lifetime fecundity was higher in the individuals that emerged from *P. persica*. Therefore, *P. persica* orchard trees were infested by *A. bungii* at much higher densities than were *C*. × *yedoensis* trees, and many adults emerged from the orchard trees. Moreover, adults that emerged from *P. persica* had higher fitness than adults that emerged from *C*. × *yedoensis*. It is clear that in Japan, *P. persica* provides more suitable food resources for *A. bungii* than does *C*. × *yedoensis*. There are three possible reasons.

The first is due to differences in the bark texture between *C*. × *yedoensis* and *P. persica*. *Aromia bungii* oviposits in cracks and crevices in the rough outer bark of trunks and large limbs. Thus, oviposition tends to be concentrated toward the lower portions of the tree in all host tree species [6,7,10,11,20,31]. This tendency is particularly strong in *C*. × *yedoensis* and young trees, which have a smooth trunk with few cracks, that are less susceptible to *A. bungii*. In *P. persica*, even 10–15-year-old trees, such as those used in this study, exhibit numerous cracks and crevices throughout the trunk that are suitable for oviposition. This is why the *C*. × *yedoensis* logs used in this study had larger diameters than the *P. persica* logs.

The second reason is that of differences in nutritional conditions for larvae between the two tree species. *Cerasus* × *yedoensis* is mainly planted for greening and ornamental purposes on roadsides and in school yards and parks, whereas *P. persica* is cultivated for agricultural production. As such, *P. persica* is more likely than *C*. × *yedoensis* to be fertilized also in the study site. Studies on Lepidopteran and Hemipteran pests that infest rice [32,33], wheat [34], tomato [35], cabbage [36], and orchards [37], as well as studies of *Hylotrupes bajulus* L. (Cerambycidae), which infests *Pinus sylvestris* [38], have found that the nitrogen contained in fertilizer accelerates the growth of pests and increases their numbers. We found that more *A. bungii* larvae bored into, and developed in, *P. persica* trunks than in *C*. × *yedoensis* trunks. When insect density is high, the amount of nutrients available to each individual is small and, consequently, emerging adults are also expected to be small. However, while 10-times more adults emerged from *P. persica* than from *C.* × *yedoensis*, the average size of adults did not differ between the two tree species. This is because *P. persica* offers good nutritional conditions due to the effect of fertilization. Cherry trees in orchards, such as *Prunus avium* (L.) and *P. pseudocerasus* Lindl., can also be affected by *A. bungii* [1]. Should these trees become infested, larval densities are expected to reach nearly the same level as observed in *P. persica*.

Finally, there are differences in the site conditions in which the two tree species occur. *Cerasus × yedoensis* is cultivated mostly in rows along roads and rivers. By contrast, *P. persica* trees are cultivated in orchards and are arranged two-dimensionally. Females of *A. bungii* can oviposit a few days after emergence, and are likely to concentrate on ovipositing near the emergence site rather than immediately dispersing. Because distances between host trees are small in peach orchards, *P. persica* is susceptible to intensive ovipositing within a short period.

Our study revealed that *P. persica* orchards pose a great risk as a source of *A. bungii*. To date, *A. bungii* has mostly affected flowering cherry trees in Japan, primarily *C*. × *yedoensis*, and particularly in the Kanto region. Flowering cherry trees are planted throughout the flat areas of Japan, a factor that has accelerated the establishment of *A. bungii* [39]. There are few *P. persica*-producing districts in the currently affected area in the Kanto region, except for Sano City, where the sample logs were collected. However, Yamanashi, Fukushima, and Nagano Prefectures, the primary *P. persica* producing areas, are adjacent to the Kanto region. Moreover, cherry (*P. avium* and *P. pseudocerasus*) production is concentrated in the Tohoku region. Once *A. bungii* invades these areas, it is expected that the large numbers of adults emerging within a short period will be difficult to control, and may seriously damage agricultural production. Early measures are required to prevent further expansion of the distributional area of *A. bungii*.

## Figures and Tables

**Figure 1 insects-13-00151-f001:**
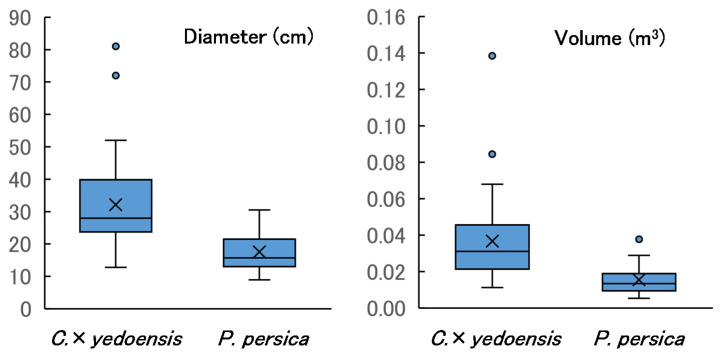
Diameter, length, and volume of sample logs. Cross marks indicate mean values. Sample sizes (*n*) were 86 (for *C*. × *yedoensis*) and 39 (for *P. persica*).

**Figure 2 insects-13-00151-f002:**
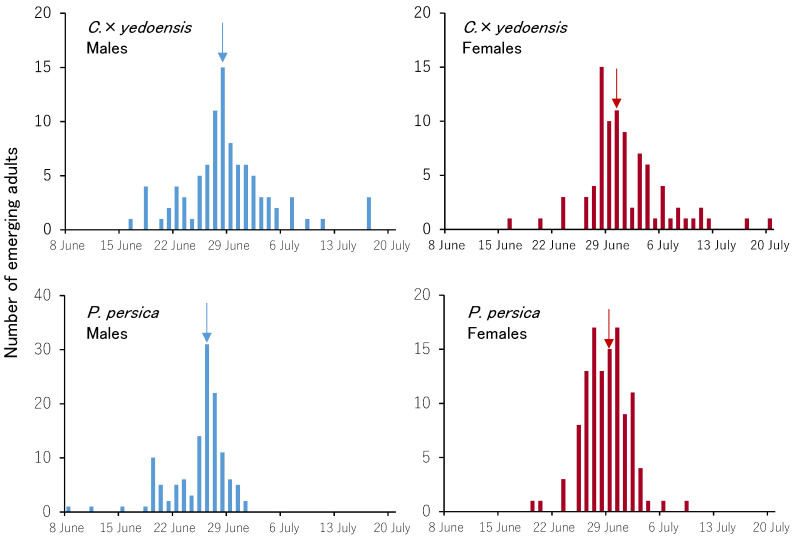
Emergence trends of *A. bungii* by sex and host species. Arrows indicate the date of 50% cumulative adult emergence.

**Figure 3 insects-13-00151-f003:**
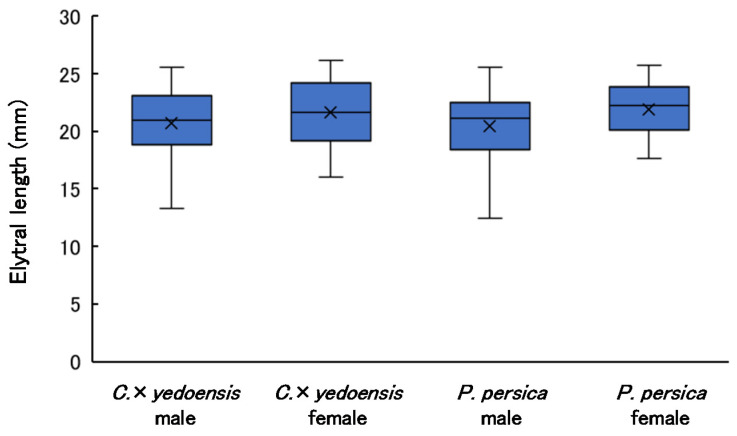
Elytral length of *A. bungii* by sex and host species. Cross marks indicate mean values. Sample sizes (*n*) were 57 (males) and 53 (females) for *C*. × *yedoensis*, and 97 (males) and 62 (females) for *P*. *persica*.

**Figure 4 insects-13-00151-f004:**
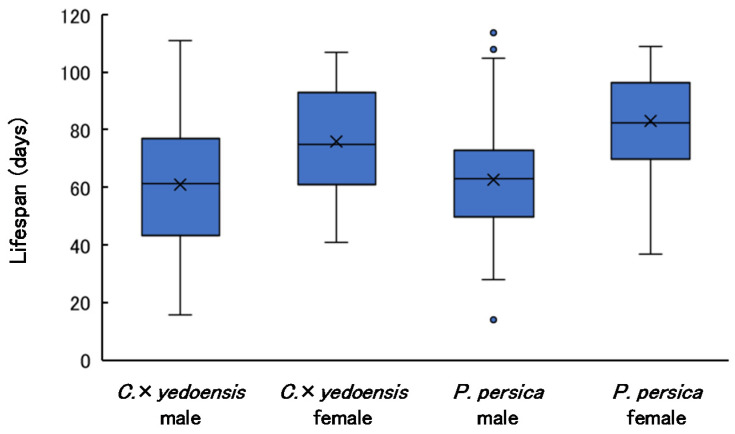
Lifespan of *A. bungii* by sex and host species. Cross marks indicate mean values. Sample sizes (*n*) were 34 (males) and 35 (females) for *C*. × *yedoensis*, and 69 (males) and 40 (females) for *P. persica*.

**Figure 5 insects-13-00151-f005:**
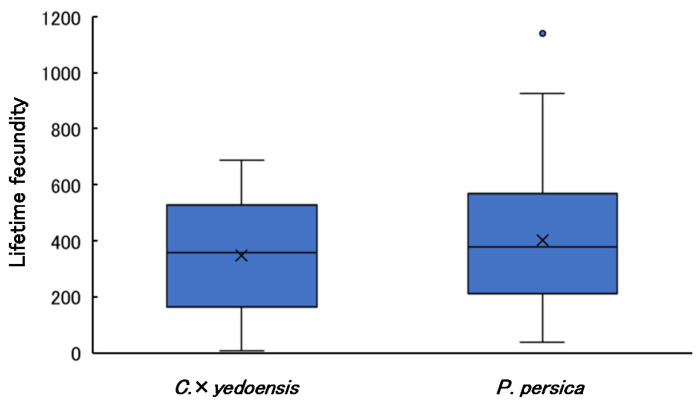
Lifetime fecundity of *A. bungii* females by host species. Cross marks indicate mean values. Sample sizes (*n*) were 41 for *C*. × *yedoensis* and 32 for *P. persica*.

**Figure 6 insects-13-00151-f006:**
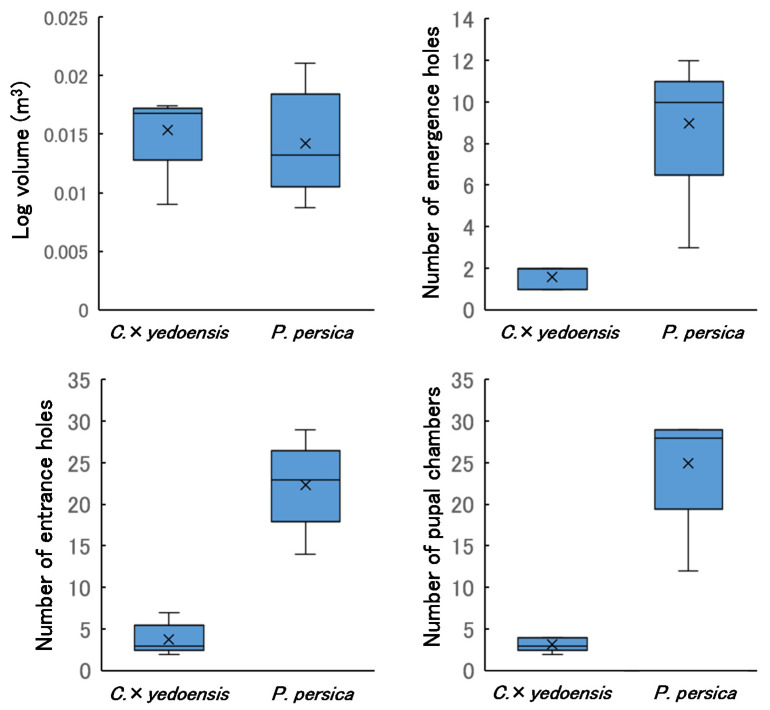
Volume of sample logs and numbers of emergence holes, entrance holes, and pupal chambers observed during log dissection. Cross marks indicate mean values. Sample size (*n*) was 5 for each tree species.

**Table 1 insects-13-00151-t001:** Number and sex ratio of *A. bungii* adults emerged from the two host tree species.

	*C*. × *yedoensis*	*P. persica*
Total tree volume (m^3^)	3.32	0.60
Number of males	70	126
Number of females	64	115
Number of males/volume	21.07	208.34
Number of females/volume	19.26	190.15
Proportion of males	0.52	0.52

## Data Availability

All data analyzed in this study are included in this article and Appendix A.

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
