# Peer review of "Comparison of the Ecological Traits and Boring Densities of Aromia bungii (Faldermann, 1835) (Coleoptera: Cerambycidae) in Two Host Tree Species"

_insects, 2022, doi:10.3390/insects13020151_

Round 1

Reviewer 1 Report

Review of Tadahisa et al. “Comparison of the ecological traits and boring densities of Aromia bungii (Faldermann) (Coleoptera: Cerambycidae) in two host tree species”

This is an interesting study on the red-necked longhorn beetle Aromia bungii, an invasive, wood-boring pest that has an increasing importance in the agriculture/forestry due to the increase of its distribution range. The data are interesting and worth publishing, however, the authors should better formulate the aim of their work and assumed hypothesis, as well as consider/explain whether the insolation of the stands (and the related temperature inside the wood) might have influenced the breeding results. To assess the suitability of a given tree species as a host plant for an insect species, samples should be ideally taken from at least 2–3 different sites for each tree species, in order to eliminate or reduce the possible impact of the quality of the habitat.

Specific comments:

Line 3: Please provide full taxonomic authority: (Faldermann, 1835).

Lines 12/13 (and further through the text): If recommended by the journal, please provide the abbreviated taxonomic authority for plant species, e.g., Prunus persica (L.), each time the name appears for the first time in the text (or separately for the main part of the work starting from the Introduction).

Line 21: Shouldn’t “to date” be placed earlier in the sentence, i.e. “have been to date more severely affected by A. bungii”?

Lines 36/37: Keywords (I) should not duplicate words already used in the title (e.g., Aromia bungii), and (II) should be arranged in alphabetical order.

Line 40: Perhaps an official abbreviation “(RLB)” could be added after “red-necked longhorn beetle”. Also here full taxonomic authority should be stated (while there is no even the abbreviated form).

Line 41: Please add some information about the distribution of this species and its increasing range.

Lines 68–79: The aim of the study is not clearly stated, and the possible impact of the results is scarcely mentioned. What was the null hypothesis?

Line 82: Perhaps a general number of investigated trees could be presented here to give a reader overall idea on a scale of the study?

Lines 84–95: Here I found the most serious flaw of the study. How about sun exposure of these sites? Don't you consider that different insolation level, and, consequently, different temperature may affect the number of emerged beetles and other characteristics of those populations? I suggest to provide some characteristic of both sites and preferably to present a photo of each habitat. Also, a few comments on possible biases connected to the issue of selecting only a single site for each tree species seems advisable.

Line 247: “one live larva” – I guess there should be “one living larva”.

Line 256: “ratio (proportion of males)” ... to females?

Line 279: “from a minimum of 10 to a maximum of 1,142” – please provide analogous data from the literature, e.g., Russo et al. 2020 that you already cited in the MS.

Lines 315/316: “Hylotrupes bajulus L. (Cerambycidae), which infests Pinus sylvestris [34]” – as far as I know Hylotrupes bajulus is a general pest of conifers and has no preference for pine (Pinus). If any, I would say it is spruce (Picea).

Author Response

Dear Reviewer 1,

Thank you very much for your reviewing of our paper. And also, we thank for your many meaningful comments. We will submit corrected manuscript with explanation for the revised points as follows. We had English proofread the manuscript again after the correction. Thank you very much for your reviewing again.

L3: We changed the description as suggested.

L12-13 (and further through the text): We provided the abbreviated taxonomic authority in the main part of the work starting from the Introduction as suggested.

L21: This sentence has already been proofread in English, so we left it as it is.

L36-37: We corrected the keywords as suggested.

L40: We changed the description as suggested (L41 in revised manuscript).

L41: We added a sentence about the distribution of this species and its increasing range (L43).

L68-79: At the end of the introduction, we added a sentence about the aim of the study (L89).

L82: We added the number of trees at L102 and L106 as suggested.

L84–95: Thank you very much for the important suggestion. Both sites were open and sunny, and it was unlikely that there would be a crucial difference in insolation. We would like to take solar radiation into consideration when conducting similar surveys in the future. Although we agree that 2 or 3 sites were need for each tree species, we could not set multiple sites at once due to a law (Invasive Alien Species Act) that restricts the transfer and breeding of A. bungii.

L247: "Live" has the same meaning as "living" and it has been proofread in English, so I left it as it is (L267).

L256: Proportion of males means the ratio of the number of males to the total number. We left it as it is because this expression is used in many literatures (L283).

L279: We added the other data about the individual difference in the lifetime fecundity (L308) as suggested.

L315/316: In addition to the cited paper, there was an internet site which shows that Hylotrupes bajulus infests the pine trees.  (https://pir.sa.gov.au/_data/assets/pdf_file/0003/295815/europeanhouseborer_factsheet19.pdf).

Sincerely yours,

Tadahisa Urano

Reviewer 2 Report

This nice manuscript describes the effects of being reared on two host species on the fitness of the invasive pest insect Aromia bungii.  The authors reared adults out of logs of the two host species, collected from field sites, and then compared a number of life history parameters. Overall, the manuscript is very clean and easy to read, and the data are nicely presented and analyzed. I have only a few suggestions for points that the authors might want to consider, as follows:

  1. Section 2.2, please describe the rearing/emergence cage conditions more explicitly. That is, the authors state that the logs were placed in large emergence cages, presumably in a big pile based on the size of the cages, and that adults were collected daily as they emerged. However, in the text, the authors should describe how exactly they collected the adults, to ensure that they collected them on the day of or soon after emergence, i.e., did the adults move to the walls or top of the cage so they were easily found, or did the authors carefully examine each log every day to detect adults hiding in the logs, or….?
  2. Throughout, in the text or the captions or legends of tables and figures, it would be useful to provide the numbers of replicates upon which the various means and standard errors are based.
  3. Section 3.7, dissection of logs, and the results: the authors have not explicitly stated a crucial point here, i.e., the overall survival of the larvae based on e.g., the number of entrance holes as the mature larvae bored into the sapwood, versus the number of exit holes, as the adults emerged, as a proxy for the survival of the beetles from late stage larvae to adults in the two tree species. In both species, they state that the number of live or dead larvae found when they split logs was relatively low, and that most pupal chambers were empty, which would tend to suggest that survival through pupation was high in both host species. Conversely, in both species, it appears that the number of entrance holes is about 2-3 times the number of exit holes, suggesting that there was considerable mortality during the later stages of development, but this is not really showing up in the numbers of live and dead larvae and adults found in the logs that were split. Thus, it would be useful to go over this more carefully in the text, and in particular, to see if there were any differences in survival from larvae to emerged adult between the two host species.
  4.  Section 3.7, it would also be useful to provide a detailed description of the state of the logs of the two host species, i.e., had the entire cambium and inner bark been consumed in one or both host species, so that the entire host resource had been consumed? It would also be useful to describe how the larvae attack the host, i.e., do the females lay egg masses, creating a starburst type pattern of larval tunnels moving outward from the oviposition site, or do the females lay eggs singly, so that each larval tunnel is separate?
  5. Discussion, L 259-269, the authors may want to explicitly state that the apparent extended emergence period of the beetles from C. x yedoensis may be an artefact of the experiment, i.e., because the logs were collected from the field, the authors had no control over when eggs were laid, and it may have been that the eggs were laid over a more extended period in this species than the P. persica.
  6. L 282-3: Because all beetles had the same oviposition conditions, is this argument really valid?
  7. L 300-301: This statement seems to be an over-simplification because the experiment used entirely field collected logs, i.e., they did not do direct comparative experiments in which female beetles were presented with host logs in choice or no choice situations under standardized conditions, and then the numbers of ovipositions, hatches, developing larvae, and emerging adults were compared. As part of this, the authors may also want to discuss the amount of host resource in the two host species, i.e., the thickness of the cambium/inner bark where much of the larval development likely takes place (?). Was this markedly different between the two species?
  8. L 310-326: do the authors know for sure that the peach trees were fertilized? If not, this argument may not be valid.
  9. The authors found that a single female beetle, fed only on honey water could lay >1100 eggs. This seems pretty remarkable, as it would constitute (just guessing) about 50% or more of the female’s body weight. Thus, do the authors have any feel for whether these numbers may have been due to the females getting nitrogen-rich nutrition from the honey water that they were fed, allowing them to develop large numbers of eggs, or is this truly reflective of the egg complement that females may have when they emerge.  Whatever the case, given that females laid >400 eggs on average, the authors may want to emphasize this in the discussion, as one reason that the beetle is able to cause rapid, extensive damage, and expand its range quickly, given this intrinsic high rate of reproduction.
  10. The authors have emphasized the value of peach as an agricultural crop, but in the introduction, they may also want to emphasize the importance and value of flowering cherry in Japanese culture, i.e., my understanding is that it is an iconic species in temple gardens and other important cultural sites, and the loss of mature flowering cherry trees from those sites would have a dramatic effect on the beauty of those sites.

Overall, a nice paper and a nice read.

Author Response

Dear Reviewer 2,

Thank you very much for your reviewing of our paper. And also, we thank for your many meaningful comments. We will submit corrected manuscript with explanation for the revised points as follows. We had English proofread the manuscript again after the correction. Thank you very much for your reviewing again.

Section 2.2: We added that we put P. persica logs in one cage, but we used three cages for C. × yedoensis because of their size (L111 in revised manuscript). We took care not to make a big pile in one cage. Emerged adults tended to actively crawl up the walls of cages or fly upwards, rarely hiding under the logs.

We provided sample sizes in each figure legend as suggested.

Section 3.7, dissection of logs, and the results: The number of adults that actually emerged from P. persica is closer to the number of pupal chambers than the number of emergence holes. This is because P. persica logs had many cracks in the bark due to the high boring density, making it difficult to find the emergence holes. We added this to the text (L272).

Section 3.7: We added that the larval boring density was high and most of cambium inner bark was consumed in P. persica logs (L272) and female adults oviposit separately (L46).

Discussion, L 259-269: Since A. bungii takes two years from oviposition to emergence and overwinters in diapause, it may be unlikely that the oviposition time affects the emergence period.

L 282-3: As you pointed out, we decided that this part is unlikely to be the reason for individual differences in the lifetime fecundity, so we deleted this part.

L 300-301: Thank you for very important suggestion. The thickness of the inner bark is currently unknown, but in the future I would like to investigate the relationship between the bark thickness and the density and the size of infesting A. bungii.

L 310-326: We added that the peach trees were fertilized in the study site (L345).

At the time of emergence, female adults hold most of eggs in the ovaries, which are deposited in their lifetime. Honey was supposed to mainly help prolong their life.

Thank you for an important suggestion. We added a sentence about the importance of flowering cherry as suggested (L61). On the other hand, the damage to the peach has been small so far and it is not regarded as a big problem. The purpose of this study is to quantitatively show the risk of A. bungii to peach which was poorly understood (L89).

Sincerely yours,

Tadahisa Urano

Reviewer 3 Report

Dear authors 

Thank you for submiting the manuscript. Apart from several comments and suggestions inserted on the manuscript I would like to know what is the original results obtained in this work.

From the discussion all relevant results are according to already published data; sex ratio, emergence period, lifespan and infested host species.

Author Response

Dear Reviewer 3,

Thank you very much for your reviewing of our paper. And also, we thank for your many meaningful comments. We will submit corrected manuscript with explanation for the revised points as follows. We had English proofread the manuscript again after the correction. Thank you very much for your reviewing again.

First of all, the original results (novelty) of this paper is that we revealed that A. bungii adults emerged from P. persica 10 times more than C. × yedoensis, and fitness elements and emergency trends were not different between adults from two tree species. Moreover, we found that P. persica trees were infested by A. bungii at much higher densities than were C. × yedoensis. As a result, we quantitatively showed the risk of A. bungii to peach which was poorly understood.

L28: The three variables are probably correlated as suggested. But here we showed the three variables (= the number of A. bungii in three stages) to emphasize the difference in density in the log between the two tree species. We could not access mortality because these variables include two generations and it was difficult to separate them.

L33 We changed “colonnade” to “street trees”.

L62-65, 67: We changed the description as suggested (L71 in revised manuscript).

L70: We added two references as suggested (L81).

L82, 90, 94: We added the number of trees as suggested (L102, 106).

L91: We used the lower part of the trunk and part of the thick branches that connected to it for C. × yedoensis. For P. persica, we used part of the trunk, which was often branching from near the ground.

L98: Thank you very much for an important suggestion. Since the logs used in the research were not treated separately for each tree, it is difficult to present the data as you pointed out.

L112 : We added that we measured the length of left elytron as suggested (L125).

L122: We changed the description. Full-grown larvae bore the outer bark from the inside to create emergence holes to facilitate the emergence of adults before overwintering. This is because the mandibles of adults are so weak that they cannot bore the bark on their own. The larvae do not emerge from the wood. Adults emerge using both the galleries and emergence holes made by the larvae (L136)

L126, 159: Thank you very much for an important suggestion. Since the logs used in the research were not treated separately for each tree, we could not conduct analysis per tree. I would like to take them into consideration when conducting similar surveys in the future.

L130 : This “condition” had the same meaning as “the numbers of live and dead individuals”, so we deleted it

Figure 1: We deleted the data about log length as suggested.

Table 1: We could not show the values for each trunk or tree because we did not treat the sample logs separately for each tree.

L182: The emergence data here show that males tend to emerge faster than females like other cerambycid species, and that there is no clear difference in emergence trends between tree species. I didn't mention the latter in the first manuscript, so I added it in discussion (L291).

L278: We added that this difference is probably the cause of the insignificant correlation (L309).

L288: The number of adults that actually emerged from P. persica is closer to the number of pupal chambers than the number of emergence holes. This is because P. persica logs had many cracks in the bark due to the high boring density, making it difficult to find the emergence holes. We added this to the text (L272). Since each sample log was less than 1 m long, the tunnels extending from the entrance holes were often cut off at the end of the log, and many of the entrance holes, pupal chambers and emergence holes found in a single log were not linked. So we could not calculate the ratio exactly as you suggested.

Sincerely yours,

Tadahisa Urano

Round 2

Reviewer 3 Report

Dear authours, thank you for considering the comments and suggestions which have clarify all doubts about the methodology and results.

Please review comments made to the new version of the  manuscript:

Line 43 - Please consult and cite original publications that first detect the spread of the beetle in each country. 

Line 121 - elytra

Table 1 - Please include minimum, maximum diameter of infested trunk, per tree species

Line 260 - one larva alive

Line 258 - add mean infestation density per tree species based on number of pupal chambers.

Author Response

Dear Reviewer 3,

Thank you very much for your reviewing of our paper. And also, we thank for your many meaningful comments.

We explain the differences between our manuscript and some Japanese citations which we and researchers who are mentioned in acknowledgments have published.

[22] Urano and Shoda-Kagaya (see attached PDF) investigated emergence trend, elytral length, lifespan and lifetime fecundity of A. bungii emerged from C. × yedoensis logs felled in Soka City. However, these data were taken in 2015 and the sample size were much smaller than the data in our   manuscript. In this study, P. persica was not treated.

[12] Shoda-Kagaya is reviewing about spread of damage caused by A. bungii in Japan and control methods. You can access to this paper online with DOI.

[39] Shoda-Kagaya discussed the spread of damage in Japan as the title suggests. There is no duplication of contents between this paper and our manuscript.

[10] Kano et al. recorded the details of the damage to C. × yedoensis in Soka City at the time of 2014. There is no duplication of contents between this paper and our manuscript.

[14] Haruyama et al. (see attached PDF) investigated the density and size distribution of larvae in P. persica trunk. Adults were not investigated. There is no data duplication between this paper and our manuscript. In this paper, C. × yedoensis was not treated.

[17] Haruyama et al. recorded the details of the damage in peach orchards. There is no duplication of contents between this paper and our manuscript and C. × yedoensis was not treated in this paper. You can access to this paper online with DOI.

We will submit corrected manuscript with explanation for the revised points as follows.

L121 (L125): We changed the word as suggested.

Table 1: Since we did not treat sample logs separately for each tree, we could not show the values for each “trunk”. We would like to take them into consideration when conducting similar surveys in the future. Instead, you can see data for infested “logs” in the right panel of Figure 1.

L260 (L269): We changed the word as suggested.

L258 (L265): We added the number of pupal chambers per wood volume as suggested.

L303 (L313): We changed the sentence as suggested.

Sincerely yours,

Tadahisa Urano
